

# Development, physicochemical characterization and *in-vitro* biocompatibility study of dromedary camel dentine derived hydroxyapatite for bone repair

Zohaib Khurshid[1,2], Mohammed Farhan A. Alfarhan[3], Yasmin Bayan[2], Javed Mazher[4], Necdet Adanir[5], George J. Dias[6], Paul R. Cooper[2] and Jithendra Ratnayake[2]

[1] Department of Prosthodontics and Dental Implantology, College of Dentistry, King Faisal University, Al-Ahsa, Saudi Arabia
[2] Department of Oral Science, Faculty of Dentistry, University of Otago, Dunedin, New Zealand
[3] Department of Surgery, College of Medicine, King Faisal University, Al-Ahsa, Saudi Arabia
[4] Department of Physics, College of Science, King Faisal University, Al-Ahsa, Saudi Arabia
[5] Department of Restorative Dentistry, College of Dentistry, King Faisal University, Al-Ahsa, Saudi Arabia
[6] Department of Anatomy, University of Otago, Dunedin, New Zealand

Corresponding author
Zohaib Khurshid,
drzohaibkhurshid@gmail.com,
zsultan@kfu.edu.sa

## ABSTRACT

This study aimed to produce hydroxyapatite from the dentine portion of camel teeth using a defatting and deproteinizing procedure and characterize its physicochemical and biocompatibility properties. Biowaste such as waste camel teeth is a valuable source of hydroxyapatite, the main inorganic constituent of human bone and teeth which is frequently used as bone grafts in the biomedical field. Fourier Transform infrared (FTIR), and micro-Raman spectroscopy confirmed the functional groups as-sociated with hydroxyapatite. X-ray diffraction (XRD) studies showed camel dentine-derived hydroxyapatite (CDHA) corresponded with hydroxyapatite spectra. Scanning electron micros-copy (SEM) demonstrated the presence of dentinal tubules measuring from 1.69–2.91 $\mu$m. The inorganic phases of CDHA were primarily constituted of calcium and phosphorus, with trace levels of sodium, magnesium, potassium, and strontium, according to energy dispersive X-ray analysis (EDX) and inductively coupled plasma mass spectrometry (ICP-MS). After 28 days of incubation in simulated body fluid (SBF), the pH of the CDHA scaffold elevated to 9.2. *in-vitro* biocompatibility studies showed that the CDHA enabled Saos-2 cells to proliferate and express the bone marker osteonectin after 14 days of culture. For applications such as bone augmentation and filling bone gaps, CDHA offers a promising material. However, to evaluate the clinical feasibility of the CDHA, further *in-vivo* studies are required.

## INTRODUCTION

Bone is a connective tissue composed of an extracellular matrix. The organic part, which makes up 35% of the matrix, consists of collagen fibres and non-collagen fibres such as "osteocalcin, osteonectin and osteopontin, glycosaminoglycans (GAGs), lipids and plasma proteins" (*Marie, 1992*). Bone contains four different cell types: osteoblasts, bone lining cells, osteocytes, and osteoclasts (*Marie, 1992*). These cells are responsible for bone growth, function and maintenance (*Yu & Wei, 2021*). The inorganic part, which makes up 65% of the matrix, consists of hydroxyapatite (*Fiume et al., 2021*; *Chen et al., 2021*). Hydroxyapatite (HA), has a Ca/P ratio of 1.667 and it has compositional similarity to human bone and teeth and demonstrates exceptional biocompatibility (*Chen et al., 2021*). Due to properties such as osteoconductivity, bioactivity and non-toxic nature, HA is commonly used as a bone grafting material (*Öksüz, 2018*; *Zhao et al., 2021*). Hydroxyapatite (HA) has been used clinically in various forms, such as powder, blocks, granules, used as coatings for biomedical implants, and in hybrid formulations with other biomaterials for increasing their osteoconductivity, mechanical properties and handling characteristics (*Ghiasi et al., 2020*; *Khurshid et al., 2022*). However, there are some limitations of HA such as a slow rate of resorption and ability to undergo remodelling. Nevertheless, additional modifications can overcome these issues, such as incorporating ions into the HA lattice and increasing the porosity for specific requirements (*Mohd Pu'ad et al., 2019*; *Ghiasi et al., 2020*).

Bone and teeth are known to be the toughest and fully calcified tissue in the human body. HA produced from teeth has received attention recently as an emerging biomaterial (*Seo & Lee, 2008*; *Öksüz, 2018*; *Ratnayake et al., 2020*; *Doğdu et al., 2021*). The calcium phosphates found in human and animal teeth have attracted significant attention as they are relatively stable when placed clinically. Hydroxyapatite (HA) can be extracted from many biological sources (such as animal bones and teeth), synthetically generated using precipitation methods using calcium and phosphate ions, and isolated utilising bioinspired approaches (*Mohd Pu'ad et al., 2019*). Numerous studies have shown that biologically generated hydroxyapatite is a strong possibility for creating efficient and affordable xenografts for bone repair (*Gao et al., 2006*; *Londoño Restrepo et al., 2016*; *Rahavi et al., 2017*; *Ratnayake et al., 2017*; *Öksüz, 2018*; *Luthfiyah et al., 2022*). In order to produce synthetic hydroxyapatite, the composition and purity of the starting material, the Ca/P mole ratio, the pH level, and the temperature of the solution should be strictly controlled. Consequently, extraction from biowaste such as animal hard tissues has been shown to provide an economical and sustainable hydroxyapatite which can be used for biomedical applications (*Mohd Pu'ad et al., 2019*). Based on our previous study, CBHA scaffolds were shown to be structurally stable, they retained their original morphology and provided a promising bone tissue engineering material for tissue repair (*Khurshid et al., 2022*). Therefore, we conducted an *in-vitro* study on extracted hydroxyapatite from camel dentine and camel cancellous bone HA as a control group. According to our knowledge there is no study reported which has investigated the chemical and biological properties of extracted hydroxyapatite from camel dentine. Therefore, the main aim of this study was to develop a xenograft material, namely hydroxyapatite, from the dentine portion of camel (Camelus Dromedarius) teeth

using a defatting and deproteinization method (*Ratnayake et al., 2020*) and characterize the physiochemical and biological properties of the camel dentine derived hydroxyapatite.

## MATERIALS & METHODS

### Camel tooth dentine extraction and processing

The study methodology was approved by the ethical committee of the Deanship of Scientific Research (DSR), King Faisal University, Saudi Arabia (KFU-REC-2022-JAN-ETHICS451). The camel (*Camelus dromedarius*) teeth were collected from a local slaughterhouse in Al-Ahsa, Saudi Arabia. Teeth were washed using distilled water, disinfected with 5.25% sodium hypochlorite and all enamel and cementum was carefully removed using a MT plus Dental Plaster (Wet & Dry) trimmer (Renfert, Hilzingen, Germany). Using an ultrasonic cleaning device at a frequency of 40 to 60 kHz for 30 min, dental pulp tissues, lipids, and blood debris were eliminated (*Ratnayake et al., 2020*). After cleaning, the dentine was dissected into tube structures using a dental high-speed handpiece.

### Defatting and deproteinization of camel dentine

Dentine samples were then subjected to defatting and deproteinisation procedures to remove all organic matter (*Ratnayake et al., 2017*; *Ratnayake et al., 2020*). Camel bone derived hydroxyapatite scaffolds (CBHA) were used as a control in this study which was prepared according to *Khurshid et al. (2022)*. Initially, the prepared dentine was pressure cooked for two hours at 15 psi (103.4 kPa) in a conventional stainless steel pressure cooker and the water ratio was set at 50 mL of water per dentine sample, as previously described in *Ratnayake et al. (2020)*. Following that, the dentine samples were dried for two hours in a desiccator. All dentine samples were soaked in a 0.1 M sodium hydroxide (NaOH) (Sigma Aldrich, St. Louis, MO, USA) solution for 8 h (NaOH solution was changed after 4 h) at 70 °C (*Uklejewski et al., 2015*; *Tanwatana, Kiewjurat & Suttapreyasri, 2019*) to remove majority of the fat. Prior to calcination, the dentine blocks were dried for 30 min at 50 °C in the oven.

### Extraction of hydroxyapatite using low heat treatment

Thermolyne, Burlington, USA, employed low heat sintering in a muffle furnace for 8 h at 750 °C to deproteinize camel dentine. Subsequently, samples were manually grounded using a laboratory mortar and pestle at room temperature under sterile conditions before being subjected to chemical analyses. Representative samples were preserved for morphological and structural analysis.

### Physiochemical characterization of the CDHA scaffold
#### *Fourier transform infra-red spectroscopy (FTIR)*

A Fourier transform infrared spectrophotometer (Perkin Elmer spectrum 100, UK) was used to identify the functional groups contained in the camel dentine derived hydroxyapatite (CDHA) samples. The FTIR spectra for CDHA sample was conducted in the mid-infra-red region in the frequency range of 400–4,000 cm$^{-1}$.

### Micro-Raman analysis

Raman spectroscopy was used to extract data on the effect of alkaline and thermal treatment on structural and compositional changes occurring in the hydroxyapatite phase of the CDHA samples. A uniform confocal geometry of the Raman spectrometer (Horiba labRAM Evolution-2; Horiba France, Palaiseau, France) was used during the acquisition of Raman spectra deploying a fixed laser power using the microscope objective at 500 mW. The laser beam was directed from the blue emission laser (He-Cd gas laser) at a fixed excitation wavelength $\approx$ 448 nm. All the Using a 100x microscopic objective lens, Raman signals were captured through a 100 $\mu$m diameter confocal hole. The Raman data collecting and spectrum charting were both done using the Labspec-6 software (Horiba, Palaiseau, France), which was also utilized for post processing the spectrum.

### X-ray diffraction (XRD) analysis

Malvern Panalytical X1 Pert MRD system (PW3040/60) was used to analyze the CDHA's crystal structure and phase composition. Analysis was conducted in the region $10° < 2\,\theta < 70°$ using Cu-K$\alpha$ radiation. The generator was set to 40 kV and 40 mA.

### Scanning electron microscopy (SEM) analysis

SEM analysis was used to assess the morphology of the CDHA scaffolds. To reveal the dentinal tubules, representative CDHA samples were cut with a scalpel blade. For the SEM and EDX analyses, the samples were subsequently coated with gold and palladium. The scanning electron microscope (Oxford JED-2300; Jeol, Tokyo, Japan) equipped with an energy dispersive X-ray (EDX, Cambridge, UK) and set at 15 kV and 15 mA was used to investigate the morphological structure and elemental composition of the CDHA.

### Inductively coupled plasma mass spectrometry (ICPMS)

An Agilent 7500 ce quadrupole ICP-MS was used to determine the elemental composition and the calcium phosphorous molar ratio (Ca:P). The results were compared with the EDX analysis.

### Thermogravimetric analysis (TGA)

Thermogravimetric analysis was conducted to determine the CDHA samples' thermal degradation. TGA was conducted using a TGA analyser (Q50, TA instruments) within a $N_2$ atmosphere. The heating rate was 10 °C per minute, from 20 °C to 1,000 °C.

## Chemical stability and biodegradation in simulated body fluid (SBF)

Firstly, a stock solution of simulated body fluid (SBF) was made following the previously reported Kokubo et al. protocol (*Kokubo & Takadama, 2008*). A scalpel blade was used to trim the samples such that their volume and mass were comparable. Each sample's mass was noted ($n = 3$). The falcon tubes were filled with the prepared CDHA sample and 10 ml of the SBF solution. The tubes were placed in an incubator at $37 \pm 1$ °C for 1, 5, 7, 14 and 21 days. At the conclusion of each time period, the pH of the solution was measured using a pH conductometer (Ionode, Acorn Scientific Ltd., Auckland, New Zealand). Subsequently, the samples were removed from the pH solution and thoroughly dried in an incubator at $37 \pm 1$ °C for 24 h before measuring the dry weight. Weight change was calculated using

the following formula and reported as a percentage WL = (W0 − W1)/W0 × 100%. W0 and W1 denotes the weights of sample before and after immersion, respectively.

### *in-vitro* biocompatibility testing of the CDHA scaffold

Cell viability/cytotoxicity testing as well as immunocytochemical analysis was conducted on both the CDHA and CBHA samples using human osteoblast-like cells (Saos-2) (*Ratnayake et al., 2023*). The cells were purchased from the American Type Culture Collections (ATCC, Manassas, VA, USA).

### *Cell culture*

Cell culture medium was prepared with minimum essential medium alpha (MEM-alpha; Invitrogen, Auckland, New Zealand) supplemented with 10% foetal bovine serum (FBS; Thermo Fisher Scientific, Auckland, New Zealand) and 5% antibiotic-antimycotic (Life Technologies, Auckland, New Zealand). The Saos-2 cells (ATCC® HTB-85™; ATCC, Manassas, VA, USA) were cultured in 25 $cm^2$ cell culture flasks within a humidified incubator at 37 °C and 5% $CO_2$. For cell seeding onto scaffolds, once cultures in the flasks reached 70–80% confluence, all the media was removed from the flasks. The cells were then trypsinised and directly seeded onto the scaffolds. Finally, the CBHA samples were prepared using an eight mm biopsy bunch. The CDHA scaffolds were prepared to a similar size to that of CBHA cutting using a scalpel blade along the axis of the dentinal tubules (∼8 mm diameter). All dissected samples were sterilised by immersion in 70% ethanol for 30 min. Samples were then placed under UV light for another 30-min cycle and rinsed in Dulbecco's Phosphate-Buffered Saline (DPBS). Finally, the sterilised and rinsed scaffolds were placed in one mL of the prepared culture medium and equilibrated over the next 24 h in a 37 °C humidified atmosphere with 5% $CO_2$. Cells were seeded at $6 \times 10^3$ per scaffold. The seeded scaffolds were incubated for 30 min to allow for attachment to the scaffold before adding one mL of culture media to each well and returning the seeded scaffolds with media into the cell culture incubator. The media was replaced daily. In addition, due to autofluorescence associated with the CDHA scaffolds, an indirect method using the elution media of the CDHA scaffold was used for the LIVE/DEAD and immunohistochemical analysis. In brief, 15 mg of the CDHA scaffold was placed in 10 mL of culture media and incubated in a humidified incubator at 37 °C and 5% $CO_2$ for 24 h. After that, the media was sterile-filtered, and the resulting elution media was used for future analysis. In addition, $6 \times 10^3$ cells were seeded on glass coverslips and incubated for 30 min to allow the cells to attach to the glass coverslip before adding one mL of the prepared elution media. The elution media was replaced daily.

### *Cell fixing and SEM analysis*

$6 \times 10^3$ cells/scaffold was seeded and incubated for 48 h on the CDHA scaffold for SEM analysis. The cell-scaffold construct was fixed with 2.5% glutaraldehyde in 0.1 M cacodylate buffer (CB), pH 7.4, for one hour with subsequent PBS and water rinse. The scaffolds were post-fixed with 1% Osmium tetroxide to improve the conductivity. Following that, the CDHA samples were washed with de-ionised water and dehydrated for 10 min in a serially diluted ethanol solution and critical point dried in a Bal-Tec CPD030 critical point dryer

(Bal-tec, Balzers, Leichtenstein). The samples were mounted on aluminum stubs, sputter coated in an Emitech K575X sputter coater with 10 nm of gold palladium. Finally, The SEM images were captured at 15 kV accelerating voltage using a SEM (Oxford JED-2300, Jeol, Tokyo, Japan).

### Cell viability/cytotoxicity

After each time period of 24, 48 and 72 h the cell-seeded scaffolds were stained using the LIVE/DEAD® cytotoxicity kit according to the manufacturer's instructions (*Ratnayake et al., 2023*). The experiments were observed using a confocal laser scanning microscope (Carl Zeiss Micro Imaging GmbH; Carl Zeiss, Jena, Germany) in order to determine the cell viability. Using the Zen 2009 software (Carl Zeiss), images were captured during visualisation.

### MTS cell proliferation assay

To analyse the proliferation of the Saos-2 cells on both the CBHA and CDHA scaffolds ($n = 3$), MTS [3-(4,5-dimethylthiazol-2-yl)—5-(3-caebozymethoxyphenyl)—2-(4-sulphonyl)—2H-tetrazolium] assay was performed. Calorimetric measurement was conducted using a spectrophotometer (Synergy 2 Multi-Mode Microplate Reader; Biotek, Berlin, Germany) at a wavelength of 490 nm. Analyses were performed after 24, 48 and 72 h. The experiment was repeated in triplicate.

### Immunohistochemical analysis

To investigate the expression of the bone-associated protein, osteonectin, using immunofluorescence, $20 \times 10^3$ cells were seeded on to glass coverslips ($n = 3$). Once the cells were attached to the coverslip, the cells were fed with elution media supplemented with 10 nM dexamethasone, 5 mM $\beta$-glycerophosphate ($\beta$-GP) and 100 μM L-ascorbic acid 2 –phosphate supplements for up to 14 days to induce cell differentiation. After the culture period (14 days), the coverslips were fixed with 10% neutral buffered formalin for four hours and then rinsed twice with PBS. Subsequently, the coverslips were blocked with donkey serum (Sigma Aldrich, Auckland, New Zealand) and then incubated with mouse anti-osteonectin (1:500 dilution; BIoGenex, USA) overnight in a refrigerator at 4 °C. The following day, the coverslips were incubated with AlexaFluor donkey anti-mouse 488 secondary antibody (Life Technologies, NZ) with excitation/emission of 495/519 nm. 4′,6-diamidino-2-phenylindole (DAPI) (Duolink; Sigma Aldrich) was used to counterstain the coverslip and visualize the cell nucleus. Finally, coverslips were observed using an Olympus fluorescence microscope (Bahns Enterprise Microscopes, Bern, Germany). Images were captured using Q-capture 3.1.2 software.

## RESULTS

### Processing of camel dentine and weight change during dentine processing

Freshly prepared dentine disc from camel tooth requires a series of treatments for removal of the organic fats and proteins. This was achieved by pressure cooking the camel dentine samples for 2 h which removed majority of the fat and collagen. After that, the dentine

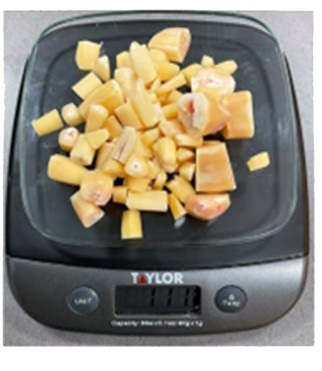
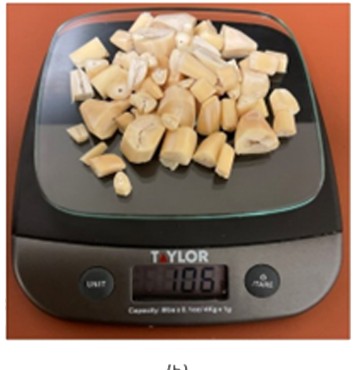
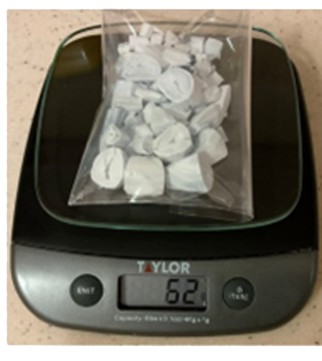

(a)  (b)  (c)

**Figure 1** **Weight changes of camel dentine after 0.1 M NaOH treatment (A, B) and 750 °C heat treatment (C).**

samples were soaked in a 0.1 M Sodium hydroxide (NaOH) (Sigma Aldrich, St. Louis, MO, USA) solution for 8 h (NaOH solution was changed after 2 h) at 70 °C which resulted in liquefied fat pouring from the dentine matrix. Weight changes indicted the removal of impurities and fats. Figure 1, shows the weight changes from (a–c) 111 g to 106 g following treatment with 0.1 M sodium hydroxide (NaOH) (Sigma Aldrich, St. Louis, MO, USA) solution for 8 h. Furthermore, a weight loss of 44% was observed after heat treatment at 750 °C.

## Physicochemical characterization of the CDHA samples
### *Fourier transform infra-red spectroscopy (FTIR)*
Figure 2 illustrates the FTIR spectra of CDHA obtained after sintering at 750 °C. Hydroxyl ($3,571$ $cm^{-1}$), phosphate ($1,092$ to $1,040$ $cm^{-1}$, $962$ $cm^{-1}$, $633$ to $566$ $cm^{-1}$ and $473$ $cm^{-1}$) and carbonate peaks ($1,455$ to $1,418$ cm $cm^{-1}$) associated with hydroxyapatite were observed for the CDHA sample, and the peaks associated with fat and collagen were not observed (*Ratnayake et al., 2017*; *Ratnayake et al., 2020*).

### *Micro-Raman spectroscopy*
Micro-Raman results for the CDHA samples indicated that CDHA is primarily comprised of HA after the defatting and deproteinising procedures. The observed HA phase was confirmed by the presence of intense Raman signature of the phosphate group ($PO_4$) arising from the stretching vibrational mode and observed in the Raman spectrum, as also shown in the Fig. 3, at $962$ $cm^{-1}$. The observed formation of the hydroxyapatite compositional phase has also been previously reported in the literature at similar spectral position of $\approx 960$ $cm^{-1}$ (*Timlin et al., 2000*; *Sofronia et al., 2014*; *Jaber, Hammood & Parvin, 2018*). The small change in the vibrational energy in our samples can be related to the higher heat treatment temperature used, which has also been previously reported and is due to the bond softening among phosphate bands at higher annealing temperatures (*Campillo et al., 2010*; *Marques et al., 2018*). The Raman peaks observed at $433$ $cm^{-1}$, $594$ $cm^{-1}$ and $1,032$ $cm^{-1}$ are also found to be directly related to the lower vibrational modes of the

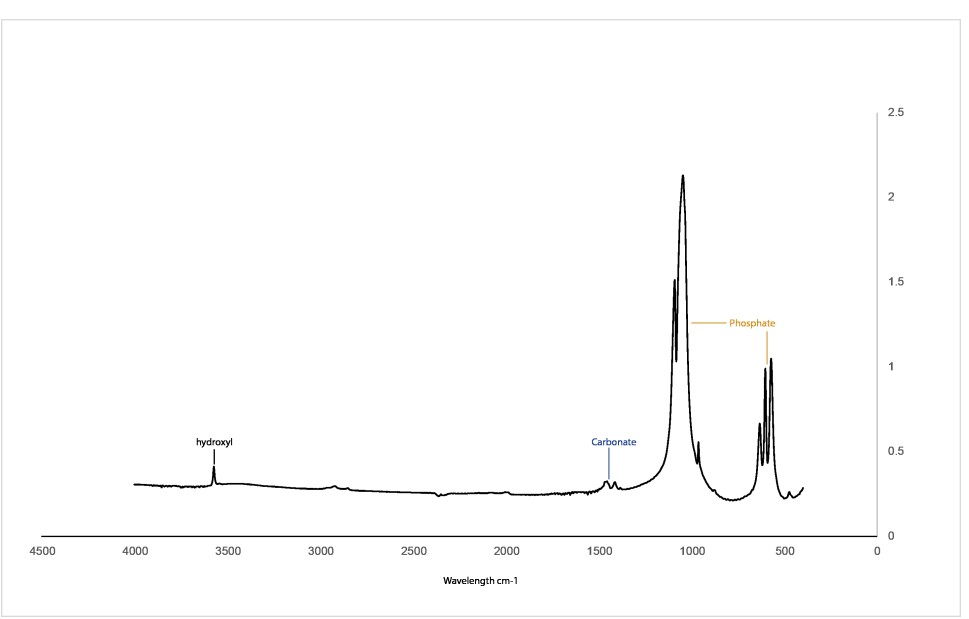

**Figure 2 FTIR spectra for the CHDA sintered at 750 °C.**

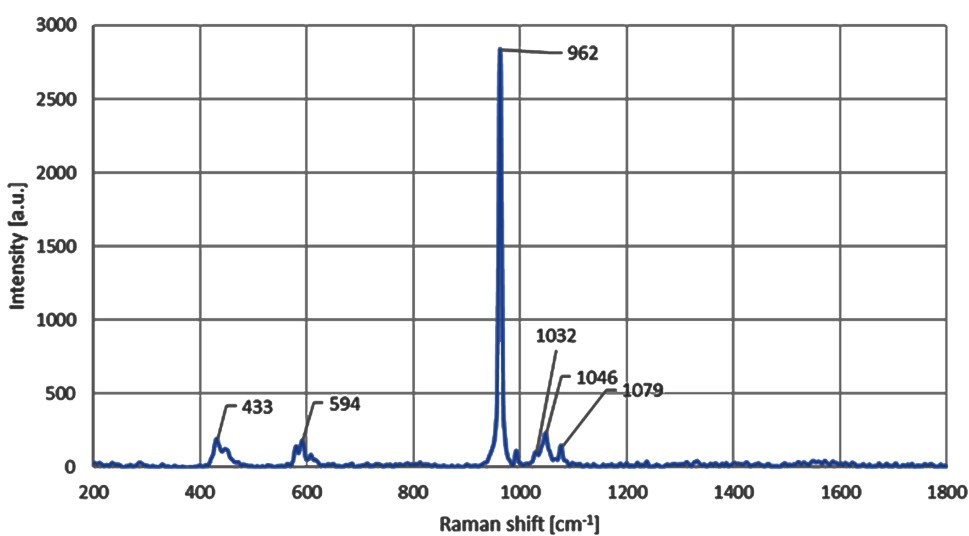

**Figure 3 Micro-Raman spectra shows the presence of peak at 962 cm⁻¹ for the CDHA sample.**

hydroxyapatite phase of the composite (*Timlin et al., 2000*; *Jaber, Hammood & Parvin, 2018*). The C-H and C-O vibrational modes have also been found in the samples in the form of peaks positioned at 1,046 cm$^{-1}$ and 1,079 cm$^{-1}$, respectively, and that has further validated the presence of HA phase in the CDHA sample (*Timlin et al., 2000*; *Londoño Restrepo et al., 2016*).

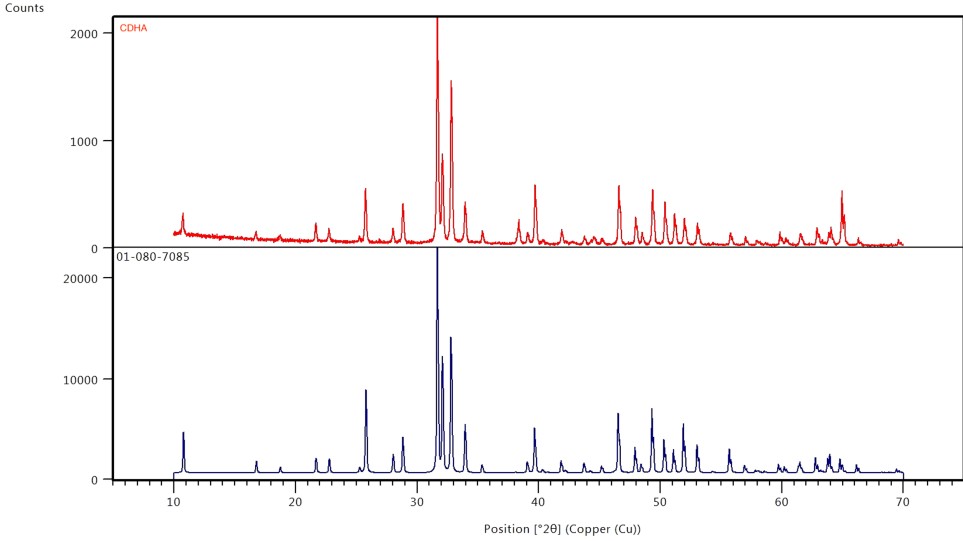

**Figure 4** X-ray diffraction patterns of CDHA (red) after sintering at 750 °C and phase pure HA (blue) (JCPDS 01-080-7085).

### X-ray diffraction (XRD) analysis

The results of the XRD analysis of the sintered CDHA powder and phase pure HA pattern (JCPDS 01-080-7085) is presented in Fig. 4. The XRD diffract grams obtained for the CDHA powder after sintering at 750 °C are very similar to the pattern for phase pure hydroxyapatite pattern (JCPDS 01-080-7085). the XRD patterns were sharper and narrower, demonstrating a high crystalline structure.

### Scanning electron microscopy (SEM) analysis

Figure 5 illustrates the microscopic porous structure of the CDHA scaffold. The presence of the dentinal tubules indicated that the fat and collagen were successfully removed from the matrix of the CDHA scaffold. The dentinal tubules (Blue arrows), measured between 1.69–2.91 μm in diameter (Fig. 5). However, some structural deterioration and debris were observed in this particular specimen (yellow arrows).

### Energy dispersive x-ray (EDX) and inductively coupled plasma mass spectrometry (ICPMS) analysis

Figure 6 shows the elemental composition of the CDHA sample obtained by EDX analysis. The inorganic component of the CDHA scaffold mainly consists of calcium and phosphate with trace amounts of sodium, magnesium and strontium.

The chemical composition of the CDHA sample analysed using ICPMS and the Ca/P mole ratio is summarized in Table 1. All values are reported in mg/kg. The ICPMS analysis further showed that toxic elements such as cadmium, arsenic and lead are lower than the concentration limits suggested by American Society for Testing and Materials (ASTM) standards (F1185-03).

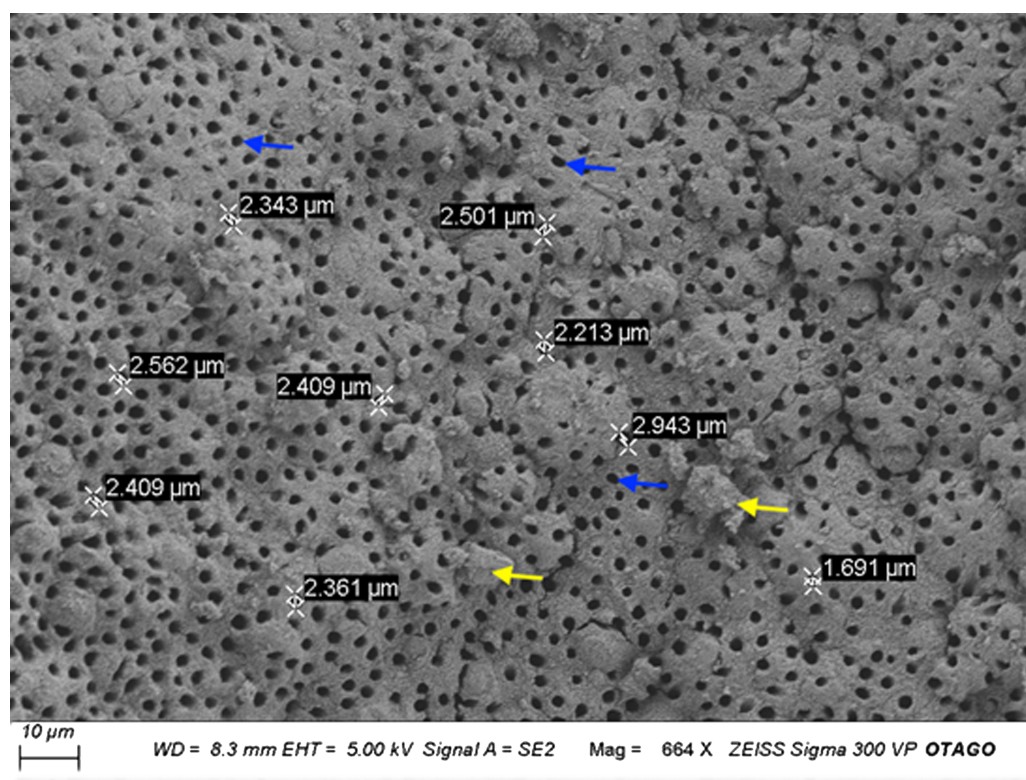

**Figure 5** **Representative scanning electron micrograph (SEM) of the CDHA scaffold.** Scale Bar = 10 μm. Blue arrows indicate dentinal tubules. Yellow arrows indicate the debris.

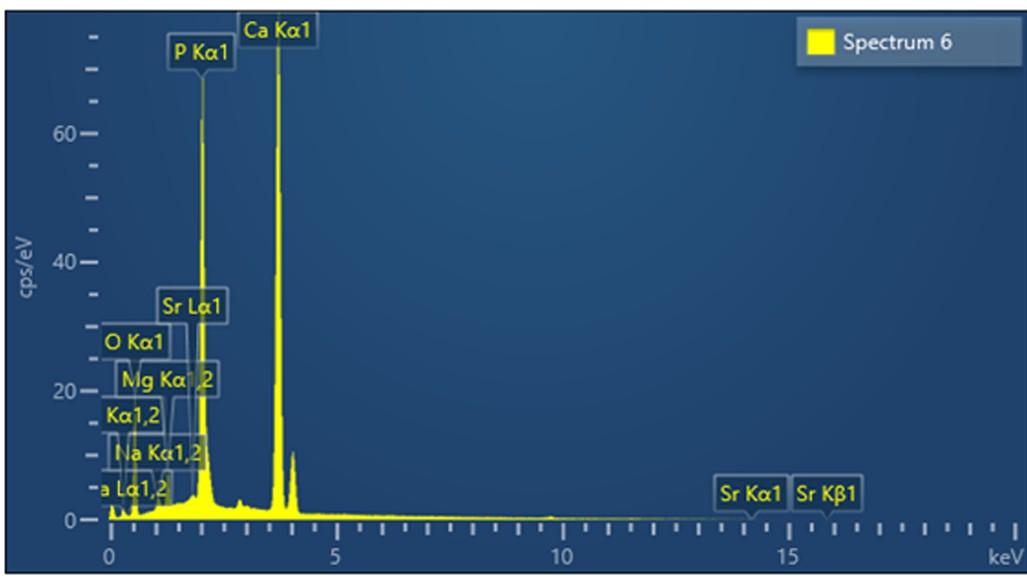

**Figure 6** **EDX analysis of the CDHA sample.**

**Table 1 Chemical composition of CDHA as determined by ICP-MS analysis and EDX analysis.** ICP-MS analysis of the CDHA sample.

| Sample (mg/Kg) | Na | Mg | Ca | P | Zn | K | Sr | Cd | As | Pb |
|---|---|---|---|---|---|---|---|---|---|---|
| CDHA | $1.15 \times 10^4$ | $2.01 \times 10^4$ | $3.94 \times 10^5$ | $1.8 \times 10^5$ | 207 | 325 | 894 | <0.25 | <0.25 | 0.35 |
| ASTM maximum limit (F1185-03) | | | | | | | | 5 | 211 | 5 |

Ca/P ratio (SEM-EDX) = 1.74
Ca/P ratio (ICP-MS) = 1.69

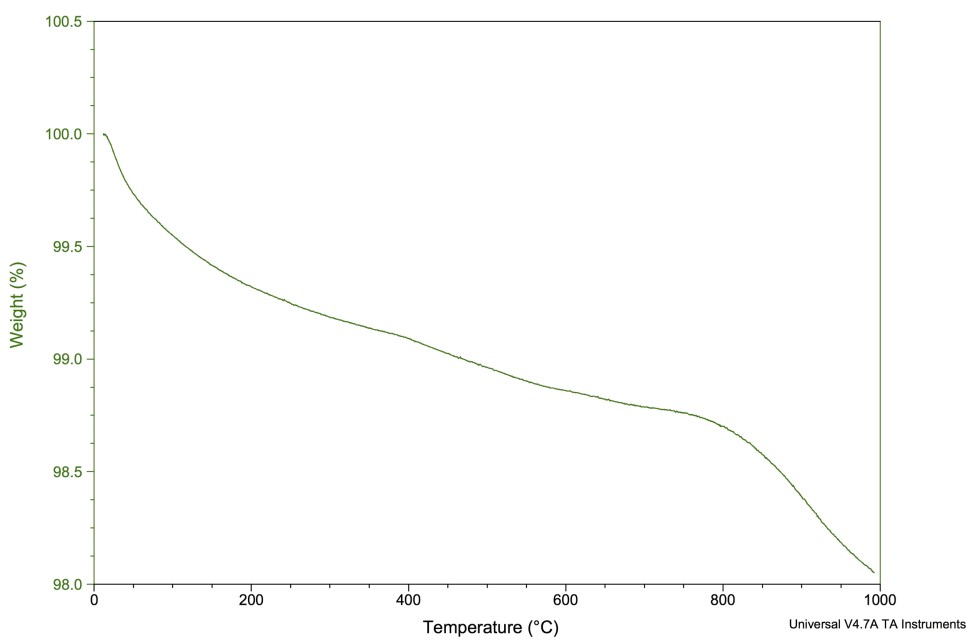

**Figure 7 TGA of the CDHA scaffold to 1,000 °C.**

### Thermogravimetric analysis (TGA)

Figure 7 shows the TGA for the CDHA when heated from 0 °C to 1,000 °C. Approximately 2% of the weight was lost from the CDHA sample when the temperature had reached 1,000 °C indicating excellent thermal stability upon sintering.

## Chemical stability and biodegradation in simulated body fluid
### Chemical stability

The initial pH of the SBF solution was 7.42. The pH of the buffer medium without the sample, which acts as a control, ranged between 7.4–7.5 throughout the experimental period. However, after day 1, the pH of the SBF solution containing the CDHA scaffold reached 9.6 by day 5 (* $p < 0.05$) and decreased to 9.2 after 21 days (** $p < 0.01$) (Fig. 8).

### Degradation

The CDHA scaffold maintained its original morphology over 21 days when immersed in SBF. Although the scaffold exhibited increased weight loss during the time period

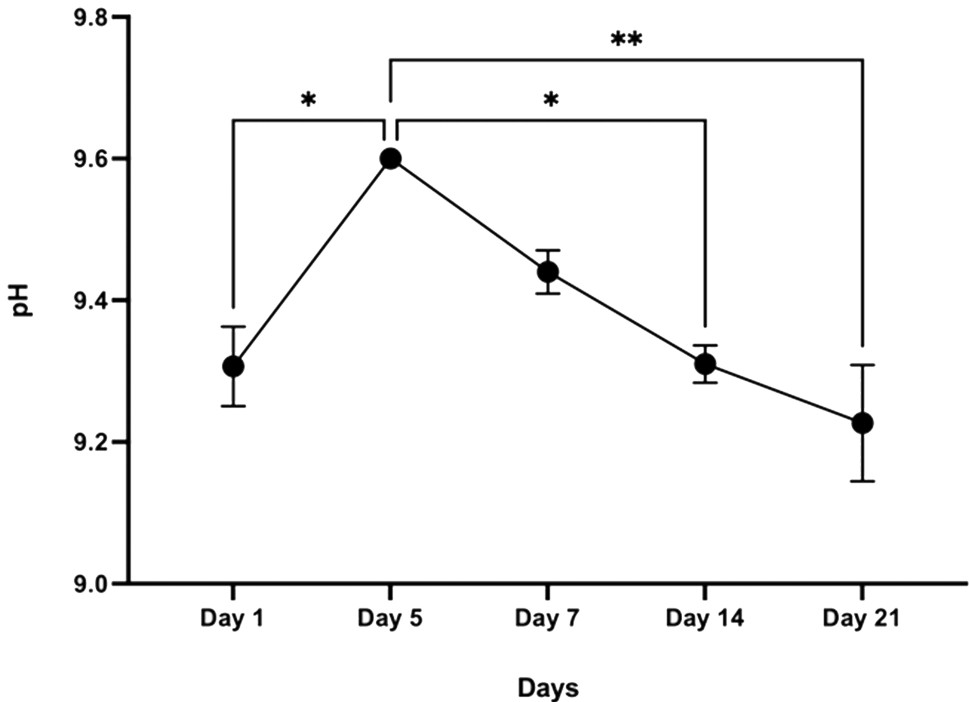

**Figure 8** **Graphical representation of pH change over time following incubation in SBF ($n = 3$).** Error bars represent $\pm$ SE of the mean after 1-way ANOVA with Tukey's multiple comparison (* $p < 0.05$, ** $p < 0.001$).

investigated, only a minimal weight loss was observed from day 1 to day 21 (~1.4%) and this result was statistically significant (**** $p < 0.0001$) Fig. 9.

### *in-vitro* biocompatibility testing
### Cell fixing and SEM analysis
Figure 10 shows an SEM image of the cellular attachment of the Saos-2 cells on the CDHA and CBHA scaffolds after 72 h culture. The cells seeded on both scaffolds exhibited slender cytoplasmic extensions (red arrows).

### LIVE/DEAD assay
Figure 11 shows the CDHA and CBHA scaffolds seeded with Saos-2 cells after performing the LIVE/DEAD Assay at 24, 48 and 72 h. Results indicated excellent cell viability for both types of scaffolds for all time periods investigated. The cells seeded on glass coverslips using the elution media of the CDHA scaffold exhibited a spindle-like morphology.

### MTS cell proliferation assay
The CDHA and CBHA scaffolds cell proliferation was investigated using the MTS assay (Fig. 12). Both scaffold types showed an increase in cell density over time. Despite the fact that the CDHA scaffold had numerically more cells than the CBHA scaffold, there was no statistically significant difference at any of the time points examined.

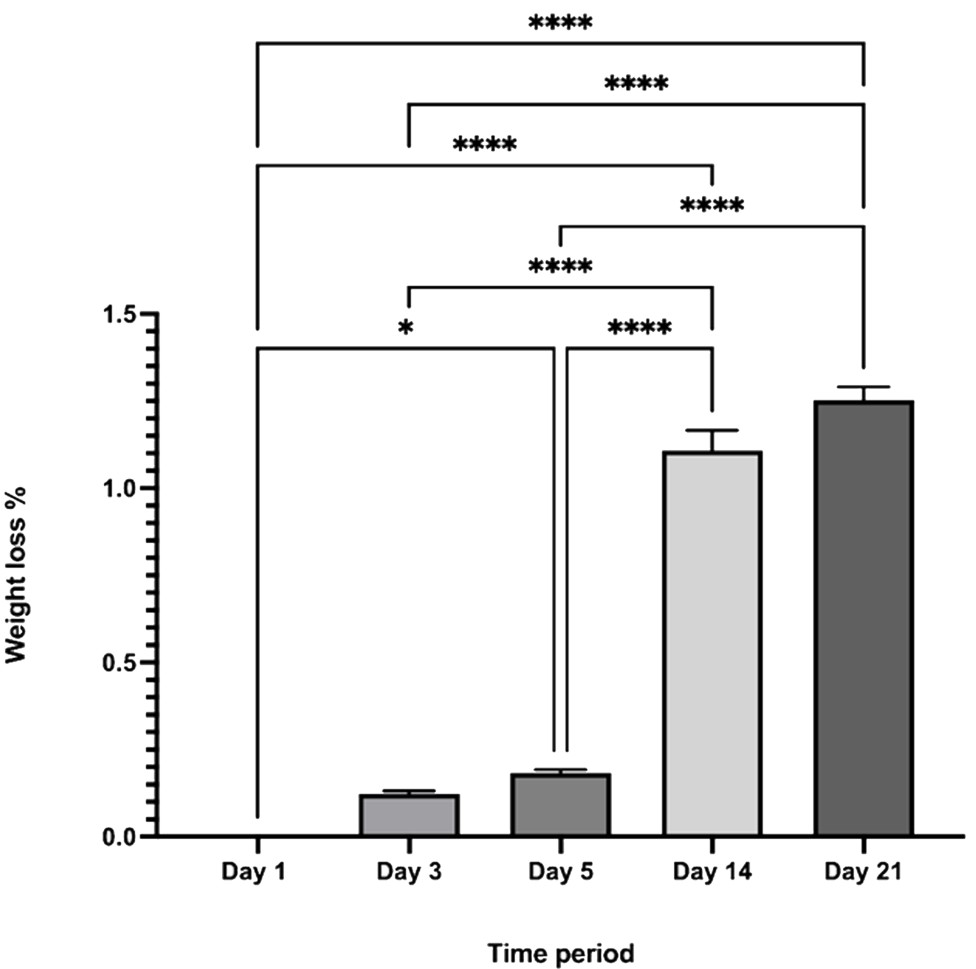

**Figure 9** **The percentage weight loss of the CDHA scaffolds *vs.* the investigated time period.** ($n = 3$. * $p < 0.05$, **** $p < 0.0001$, error bars represent +SE of the mean after one-way ANOVA with Tukey's multiple comparison).

### *Immunohistochemical analysis*

Immunohistochemical analysis showed that the Saos-2 cells expressed the bone marker osteonectin on the surface of the CDHA scaffold after 14 days culture (Fig. 13).

## DISCUSSION

A xenograft material, hydroxyapatite, was successfully produced from the dentine portion of camel teeth using a simple, cost-effective and reproducible defatting and deproteinisation method (*Ratnayake et al., 2017*; *Ratnayake et al., 2020*; *Khurshid et al., 2022*). Pressure cooking is an efficient method of removing fat and collagen from the dentine matrix as it denatures collagen. NaOH acts as a detergent removing fats and proteins. Fat absorbs microwave energy faster and is a cost-effective method of removing fat. It has previously been reported on the effectiveness of heat-treatment for deproteinization of bone cubes or other animal biowaste for use in osseous healing, such as bone graft and voids filling

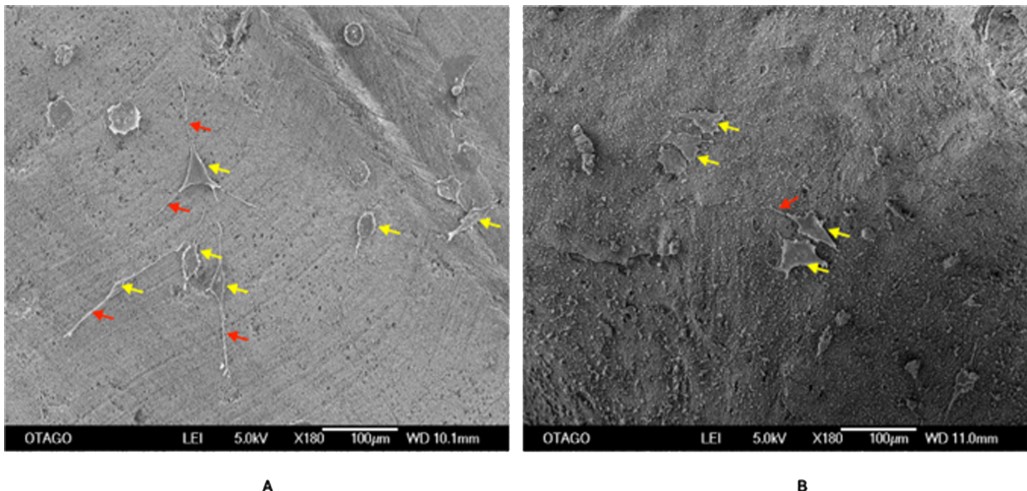

**Figure 10** **SEM image of the CDHA (A) and CBHA (B) scaffold with attached Saos-2 cells after seeding for 72 h.** Yellow arrows indicate Saos-2 cells. Red arrows indicate cytoplasmic projections. Bar = 100 μm.

(*Khurshid et al., 2022*). Consequently, the samples were treated at 750 °C for 8 h in a dry heat furnace. It was evident that mass and colour changes (from yellow into chalky white) of the camel dentine samples subsequently occurred (Fig. 1).

The FTIR spectra of the CDHA showed the characteristic peaks associated with hydroxyapatite (Fig. 2) (*Joschek et al., 2000*). In this study, peaks associated with collagen and fat were not observed, suggesting that all the organic matter was removed (*Barakat et al., 2009*). The sharp peak observed at 3,571 cm$^{-1}$ is due to the presence of the hydroxyl group. The peaks observed at 1,092 to 1,040 cm$^{-1}$, 962 cm$^{-1}$, 633 to 566 cm$^{-1}$ and 473 cm$^{-1}$ represent the phosphate bands (*Elkayar, Elshazly & Assaad, 2009*; *Ramesh, Ratnayake & Dias, 2021*). In addition, carbonate was retained within the CDHA structure, which is confirmed by the carbonate bands observed at 1,455 to 1,418 cm cm$^{-1}$ and 873 cm$^{-1}$, respectively. Similar studies which used mammalian sources to produce hydroxyapatite, bone matrix-associated phosphate, carbonate and hydroxyl peaks were also observed in their respective FTIR spectra (*Ratnayake et al., 2017*; *Ratnayake et al., 2020*; *Khurshid et al., 2022*).

Raman analysis was performed on the CDHA sample and the spectral positions of the Raman peaks suggest a strong presence of hydroxyapatite phase owing to the observation of all the four characteristic vibrational modes of phosphate bands in the CDHA sample (*Timlin et al., 2000*; *Sofronia et al., 2014*; *Jaber, Hammood & Parvin, 2018*). Furthermore, all the four major Raman vibrational modes of the sample were also compared with the high quality spectral data obtained from the RRUFF hydroxyapatite database (ID number R100225) (*Lafuente et al., 2016*). A perfect Raman peak matching was observed for all the four peaks with that of the $Ca_5(PO_4)_3OH$'s reference Raman database shown in Fig. 3. Furthermore, the presence of 1,032 cm$^{-1}$ Raman mode in our sample indicates a significant degree of stoichiometric retention in the hydroxyapatite phase during the chemical and thermal treatments of the CDHA samples (*Timlin et al., 2000*). The retention mode ≈

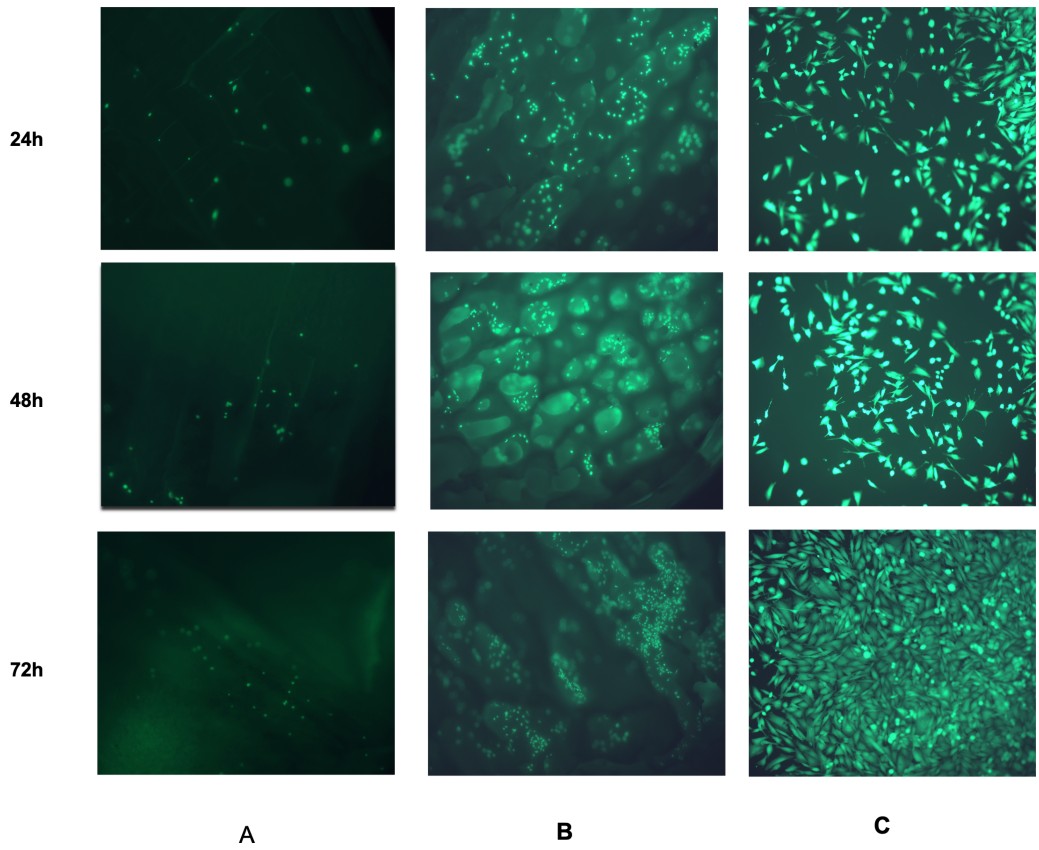

**Figure 11 Fluorescent images from the LIVE/DEAD® assay of Saos-2 cells growing on CDHA scaffolds.** Image shows cell viability at 24 h, 48 h and 72 h culture. (A) Saos-2 cells seeded directly on the CDHA scaffold. (B) Saos-2 cells seeded directly on CBHA scaffold. (C) Saos-2 cells seeded with the elution media of the CDHA scaffolds. Green, live cells (calcein), Red, dead cells (ethidium homodimer-1).

1,032 cm$^{-1}$ has been reported to be Raman inactive in the non-stoichiometric HA phase (*Timlin et al., 2000*).

The XRD diffractograms showed sharp and narrow peaks indicating increase crystallinity and corresponded to the peaks associated with synthetic hydroxyapatite (Fig. 4). In addition, secondary phases such as calcium oxide ($2\theta = 37°$) and tricalcium phosphate ($\beta$-TCP) ($2\theta \approx 31.13°$) were not observed suggesting that the CDHA has undergone minimal decomposition after sintering at 750 °C leaving the carbonate in the lattice which was further confirmed by the FTIR spectra (Fig. 2). Several other studies found similar findings when hydroxyapatite was synthesized from biowaste (*Barakat et al., 2009*).

SEM analysis of the CDHA scaffold showed an interconnected micro-porous structure consisting of dentinal tubules. These dentinal tubules ranged from 1.69–2.91 μm, and these micropores could assist in enabling angiogenesis and cellular attachment (*Timlin et al., 2000*). The debris observed (Fig. 5, yellow arrows) in the CDHA scaffold is likely due to the defatting and deproteination processes.

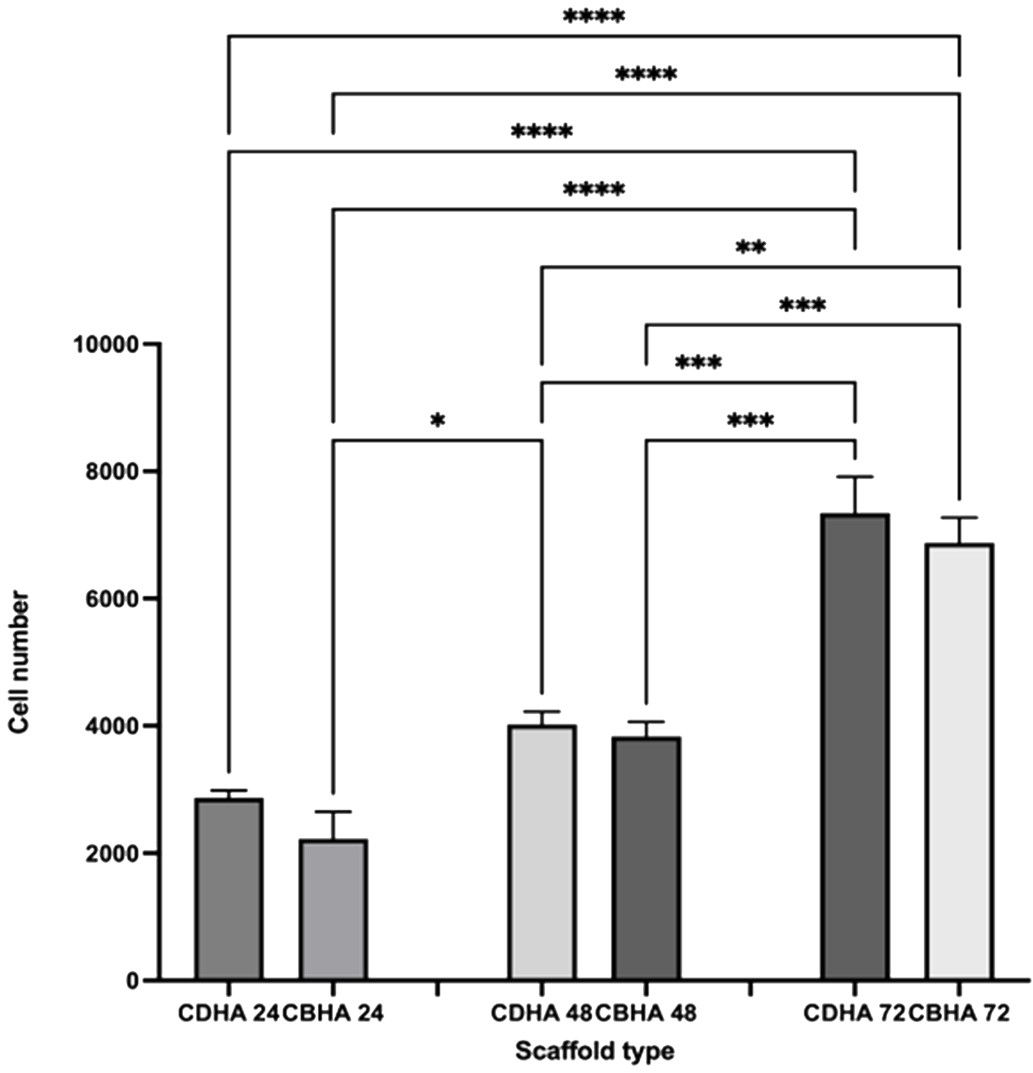

**Figure 12** **MTS cell proliferation assay on CDHA and CBHA scaffolds after 24, 48 and 72-h culture.**
Error bars represent $\pm$ SE of the mean after 1-way ANOVA with Tukey's multiple comparison test. $*p <$
0.05, $**$ $p < 0.01$, $***$ $p < 0.001$, $****$ $p < 0.0001$. $n = 3$.

The trace elements found in the CDHA scaffold could play a vital role in bone metabolism
(*Landi et al., 2008*; *Krishnamurithy et al., 2014*). Sodium plays a crucial role in cell adhesion
and the bone mineralisation process. Magnesium stimulates the proliferation of osteoblasts,
and strontium enhances osteoblast activity whilst inhibiting osteoclast activity (*Bigi et al.,
2007*; *Landi et al., 2008*). A slightly higher Ca:P mole ratio was calculated for CDHA
compared with stoichiometric HA (1.67) through EDX and ICP-MS analysis (Table 1).
The slightly higher value was due to the carbonate groups and trace elements in the CDHA
sample (*Bahrololoom et al., 2009*; *Ratnayake et al., 2017*). The result observed in this study
is similar to our previous studies which used bovine teeth and bone as a.

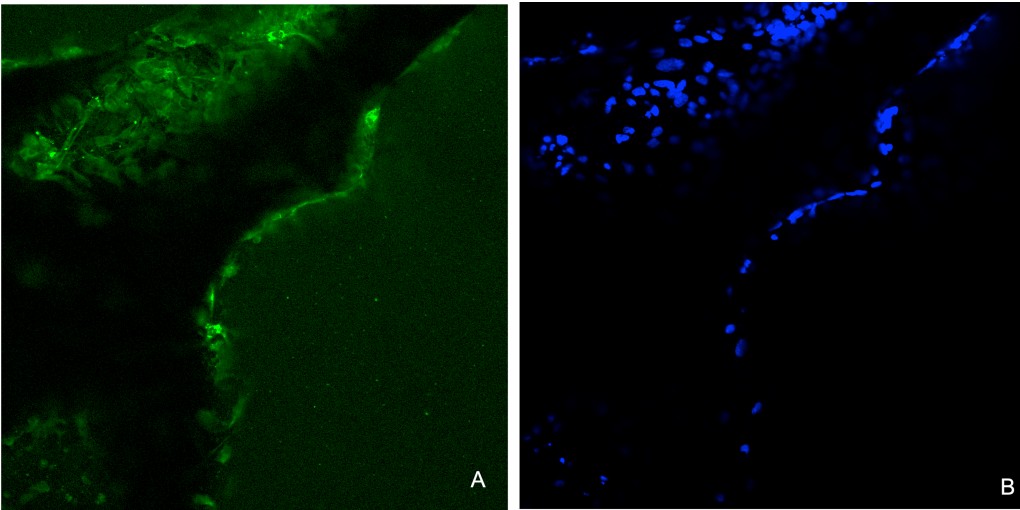

**Figure 13** **Representative images for the immunohistochemical analysis for osteonectin after 14 days on the CDHA scaffold after seeding of 20 × 103 Saos-2 cells.** (A) Positive immunofluorescence staining (green) for osteonectin. (B) Nuclear staining (DAPI staining, blue) of the Saos-2 cells.

According to the TGA analysis, the CDHA scaffold only decreased by ∼2% of the weight suggesting an excellent thermal stability for the CDHA scaffold. The initial weight loss (endothermic loss) from 50 −200 °C was attributed to the evaporation of absorbed water. The minimal weight loss (∼1%) between 250–600 °C (exothermic loss) was due to the dissociation of the carbonate groups in the CDHA lattice (*Saalfeld et al., 1994*). A similar finding was observed in our previous study which utilized the dentine portion of bovine teeth (*Ratnayake et al., 2020*).

Simulated body fluid (SBF) mimics the intracellular compartment of human body fluids and tissues (*Kokubo & Takadama, 2006*). The pH increased significantly to 9.6 by day five and dropped to 9.2 by day 21 following sample treatment. This significant increase in pH could be due to the decomposition of the carbonate groups in the CDHA into CaO which reacts with the water to form $Ca(OH)_2$. Another factor could be the leaching of $Ca^{2+}$ ions and trace amounts such as Na, Mg, K and Sr from the CDHA lattice into the SBF solution to form an alkaline solution (NaOH, $Mg(OH)_2$, KOH, $Sr(OH)_2$) (*Ratnayake et al., 2017*; *Ratnayake et al., 2020*). The degradation study showed that the CDHA scaffold decreased by ∼1.5% of its original weight after 21 days treatment. Several studies have shown minimal HA degradation (*Gomi et al., 1993*; *Monchau et al., 2002*). For example, Legeros et al. showed that physical properties such as a higher porosity and interconnected porous network increase the biodegradation of a scaffold (*LeGeros et al., 1988*). However, the CDHA scaffold exhibited a dense architecture consisting of micropores ranging from 1.69–2.91 $\mu$m, which would have minimised the biodegradation. In a previous study, a HA scaffold produced from bovine dentine showed a similar result, losing only ∼2.5% of its original weight. However, the bovine dentine-derived HA (BDHA) pore sizes were larger than in the CDHA studied here (*Ratnayake et al., 2020*). Limitations of the biodegradation

experiment included the absence of primary enzymes, such as lysozymes, responsible for the scaffold's degradation in physiological environments. In addition, the scaffold was not placed in a true physiological environment where mechanical forces of perfusion and cellular waste products would degrade the scaffold.

Human osteosarcoma (Saos-2) cells are commonly used to assess the biocompatibility of orthopedic materials due to its osteoblast like properties. Cells were seeded at $6 \times 10^3$ directly on to the surface of the scaffolds to evaluate their adhesion and proliferation. This specific cell density has been identified as reaching an appropriate confluency over the investigated time period (22 h) (*Ratnayake et al., 2020*). However, due to the autofluorescence and cells being adhered in different planes on the CDHA scaffolds, the elution media of the CDHA scaffold was used for the LIVE/DEAD assay. The SEM image (Fig. 10) showed that the cells adhered to the CDHA and CBHA scaffolds and exhibited cellular projections which indicates intracellular communication and proliferating. The LIVE/DEAD assay showed the CDHA scaffold is non-toxic, allowing cells to adhere and proliferate (Figs. 11A, 11C). In addition, no dead cells were present indicating increased growth and viability over the cultured period. Indeed, the cells penetrated deeper into the pores of the CBHA scaffold, and the cells were viable (Fig. 11B).

The MTS assay showed that the cells proliferated during the investigated time period for both the CDHA and CBHA scaffolds (Fig. 12). Although the cell numbers were marginally numerically higher for the CDHA scaffold compared with the CBHA scaffold, this result was not statistically significant. The increased cell proliferation for the type of scaffolds indicated that the cells adhered to the scaffolds and proliferated. One of the reasons for this may be due to the trace elements (Carbonate, Mg, Na, K and Sr) present in the scaffolds as well as the topography (*Parsons et al., 1988*; *Nair & Laurencin, 2007*; *Ratnayake et al., 2020*). Due to the dentinal tubules, the CDHA scaffold exhibited an uneven topography. *Deligianni et al. (2000)* and *Golub & Boesze-Battaglia (2007)* reported that a porous architecture and a rough surface promote a significantly increase in cell attachment. In addition, the LIVE/DEAD assay and SEM analysis (Fig. 10) showed that the cells exhibited cytoplasmic projections known as filopodia, which act as anchorage points for cells to facilitate cell adhesion and migration.

For immunohistochemical analysis, $20 \times 10^3$ cells were cultured on the well plate using the elution of the CDHA scaffold. An indirect method was used to eliminate autofluorescence associated with the CDHA scaffolds. A 14-day culture period was chosen according to a study Gao et al. who found that a cultivation period of 14 days was necessary to express osteonectin on a nano-porous silicon substrate (*Gao et al., 2006*). Osteonectin is a bone matrix protein associated with bone mineralization and is used as a bone marker. Immunohistochemical/immunofluorescence analysis demonstrated expression of the bone marker osteonectin after 14 days indicating *in-vitro* osteogenic differentiation leading to extracellular bone matrix formation.

## CONCLUSIONS

This study showed that waste camel teeth are a potential source to extract hydroxyapatite using a simple, cost-effective, reproducible method. Although the CDHA material showed

excellent biocompatibility properties in *in vitro*, further *in vivo* research is warranted to evaluate its feasibility as a bone substitute for clinical applications such as bioactive coatings for orthopaedic implants, bone void fillers and dental bone grafting.

## ACKNOWLEDGEMENTS

We acknowledge the College of Medicine, King Faisal University, KSA for supporting with provision of lab facilities and the Sir John Welsh Research Institute (SJWRI), University of Otago, New Zealand for helping with sample testing. Technical support from Ali Khaled Abutaleb (Dental Technician), College of Dentistry, King Faisal University, KSA is acknowledged.

### Funding

This work was supported by the Deputyship for Research & Innovation, Ministry of Education, Kingdom of Saudi Arabia (Project No. INST179). The funders had no role in study design, data collection and analysis, decision to publish, or preparation of the manuscript.

### Grant Disclosures

The following grant information was disclosed by the authors:
Deputyship for Research & Innovation, Ministry of Education, Kingdom of Saudi Arabia: INST179.

### Competing Interests

The authors declare there are no competing interests.

### Author Contributions

- Zohaib Khurshid conceived and designed the experiments, performed the experiments, analyzed the data, prepared figures and/or tables, authored or reviewed drafts of the article, and approved the final draft.
- Mohammed Farhan A Alfarhan conceived and designed the experiments, performed the experiments, analyzed the data, authored or reviewed drafts of the article, and approved the final draft.
- Yasmin Bayan performed the experiments, analyzed the data, prepared figures and/or tables, authored or reviewed drafts of the article, and approved the final draft.
- Javed Mazher performed the experiments, analyzed the data, prepared figures and/or tables, and approved the final draft.
- Necdet Adanir conceived and designed the experiments, analyzed the data, authored or reviewed drafts of the article, and approved the final draft.
- George J. Dias analyzed the data, authored or reviewed drafts of the article, and approved the final draft.

- Paul R. Cooper conceived and designed the experiments, analyzed the data, prepared figures and/or tables, authored or reviewed drafts of the article, and approved the final draft.
- Jithendra Ratnayake conceived and designed the experiments, performed the experiments, analyzed the data, prepared figures and/or tables, authored or reviewed drafts of the article, and approved the final draft.

## Animal Ethics

The following information was supplied relating to ethical approvals (i.e., approving body and any reference numbers):

KFU-REC-2022-JAN-ETHICS451

## Ethics

The following information was supplied relating to ethical approvals (i.e., approving body and any reference numbers):

This study was approved by the Research Ethics Committee of the King Faisal University, Kingdom of Saudi Arabia (KFU-REC-2022-JAN-ETHICS451).

## Data Availability

The raw data is available in the Supplementary File.

## Supplemental Information

Supplemental information for this article can be found online at http://dx.doi.org/10.7717/peerj.15711#supplemental-information.

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
