# Peer review of "Development, physicochemical characterization and in-vitro biocompatibility study of dromedary camel dentine derived hydroxyapatite for bone repair"

_PeerJ, doi:10.7717/peerj.15711_

## Round 0.1 · original submission · Major Revisions

Please address issues pointed out by all reviewers and amend the manuscript accordingly.

Reviewer 1 ·

Basic reporting

The paper is well written and very clear to understand the processes. The Author had earlier also published a couple of papers and reviews on this subject. The research topic has got novelty as they have given a detailed synthetic approach to synthesize the hydroxyapatite. The synthesized HAp was characterized with various analytical instruments to match the required specification. The research activity needs further investigation which will help in filling the bone gaps and bone augmentation applications.

The Literature references referred to were of relevant fields and have deep knowledge of research activity going on to date on this subject.

The manuscript contains all the figures, tables, and data required for judging the results.
As a reviewer, I am able to read and understand the research intent and its application in the near future.

Experimental design

The research activity is very well-defined and rigorous investigation is done with the highest technical details. The animal ethical standards were also kept.

In General, the experimental design was excellent and written. some minor changes addition, and modifications are as follows,

1. Page 8, Line 123 - FTIR - Pl. mention the model used like Spectrum 100 etc
2. Page 8, Line 131 - The Raman spectrometer, an instrument used should be written as Horiba labRAM
3. Page 8, Line140 - The XRD instrument should be written as - Malvern Panalytical X1 Pert MRD system
4. Page 8, Line 147 - The SEM instrument should be written as - JED-2300
5. Page 18, Line 534 - The year of publication is 2015
6. Page 24, Figure-2 - the FTIR Spectral data is collected in % Absorbace in Y-Axis. It would be better if we reprocess the data in % Transmission mode as all the results mentioned in the experimental observation are in % T mode.
7. The Raw data is shared as - FTIR_DCHA - The Excel file has three sheets having collected data but could not confirm which results belong to DCHA. All three sheets have data, so request you to relook into this and label the sheets appropriately.
8. The XRD raw data - Pl. name the columns C and D, the values belonging to which experimental activity.

Validity of the findings

The outcome of the results was comparable with the earlier work published, especially the Ca/P ratio.

Additional comments

This is an interesting research work. Currently, we see a lot of people, especially now kids also suffering from tooth decay issues. Bone loss is a common problem and if we get something good out of this work it would be great for our human kind.

Reviewer 2 ·

Basic reporting

This research article is certainly adding value to the process of producing of hydroxyapatite (HAp) from the camel teeth.
Since hydroxyapatite (HAp) is widely used in bone repair, replacement, and augmentation as well as tissue engineering scaffolds for bone regeneration, its easy availability to prevent shortages in its supply chain has sparked a high level of interest in conducting research.
Zohaib Khurshid and others have made concerted efforts to produce it from the dentine portion of camel teeth using a defatting and deproteinising process.

The manuscript was well writing and easy to understand, the diagram, tables and literature references are well aligned with the discussion and findings. I recommend it for publication.

Experimental design

The CDHA production method was very well explained, starting with extraction and then characterization with several advanced techniques such as X-RD, SEM-EDX, and ICP-MS. The research also studies chemical stability, biodegradation, and biocompatibility, these data are important for its commercial viability.
However, the below correction will give more clarity,
• Page 8, Line 131; please mentions the "Horiba labRAM."
• Page 8, Line 140; please mention "Malvern Panalytical X1 Pert MRD System"
• Page 8, Line 147; The SEM instrument should be written as "JED-2300."
• Page 24, Figure 2: The FTIR spectral data is having Y axis as “% Absorbance”, please use the “% Transmission”, as FTIR data was always represented in % Transmission rather than % Absorbance.

Validity of the findings

The study findings and analysis are consistent with past works, including in vitro biocompatibility testing, which revealed that the 459 CDHA scaffold was non-toxic.

Additional comments

The study has the ability to excite the new researcher's interest in conducting in-vivo investigations to evaluate the clinical feasibility of the CDHA.
The production method may be employed on a commercial scale, which will help reduce the shortage of hydroxyapatite (HAp) for human health.

Reviewer 3 ·

Basic reporting

This manuscript focuses on the production hydroxyapatite from the dentine of camel teeth. The primary aim is simply the characterization of this hydroxyapatite. The methods are thoroughly presented and range from the extraction of and sample treatments to the characterization, cell viability and immunohistochemical analyses, among others. Overall, the ideas in manuscript could use better organization and the text must be verified for clarity throughout.

Experimental design

The experimental design used is appropriate to document the production of hydroxyapatite. The multiple evidence approach used, which combines SEM, AMS, and various chemical analyses seem provide evidence for camel teeth provide a potential source of hydroxyapatite. The study is largely descriptive; thus, refining/clarifying the aims and tidying up the methods and results sections would improve readability and really make the manuscript flow better. The methods were generally well presented, although a couple clarifications are necessary.

Validity of the findings

The authors presented a highly descriptive approach to document the production of hydroxyapatite from camel teeth. The title of the study emphasizes the potential of camel dentine hydroxyapatite, but this is downplayed in the manuscript. The primary aim of the study seems to fall short, and primarily relies on documentation and description without providing a broader context of the importance of the study. The evidence presented does seem support the findings and conclusions in general. However, the while results are presented in subsections corresponding to the methods, most subsections are generally short, and several sentences are not very informative (i.e., merely repeating some methods). This makes the results section a bit distracting and difficult to go through. The discussion at times seems to lack a broader perspective in some paragraphs with very few citations. This makes it difficult to gauge the thoroughness of the review, the applicability of the methods in a broader context, and the how the presented results compare to other studies in general. Similarly, most of the text in the conclusion statement contains sentences that are generally repeated from reading other parts of the manuscript. The authors should consider working on these sections and pay close attention to the applicability of these methods and their findings in a broader sense.

Additional comments

The list below includes line comments for the authors to consider:
Title:
Not much is discussed in the manuscript about the application potential of the synthesized hydroxyapatite. Perhaps this could be reworded for clarity or rewrite sections of the manuscript to reflect more transparently the statement in the title.

Abstract:
General: include a clear aim.
L28: first sentence seems to be more appropriate as a second sentence. Similarly, this sentence could elaborate more about why camel teeth could be considered an importance source of hydroxyapatite.
L31: “approached” – use correct tense.

Introduction:
L49: “… with a more complex composition …” – this fragment is missing a comparative statement.
L55: “Other essential features of Hap …” – the use as bone graft material could be better developed.
L58: reference after Zhao et al. seems to have some odd characters.
L58: “It has been used clinically …” – unclear what “it” refers to.
L61: Reword sentence and avoid double conjunction “however”.
L62: “… by additional modification …” – this fragment here needs an example to improve clarity.
L65: remove adverb “Previously”
L74: “The composition …” – revise sentence for clarity.
L77: “provides” – use correct tense.
L82: “HA” – this was previously abbreviated as “HAp”. Pick one and change throughout.
L85: this hanging sentence should be added to the previous paragraph. Also, this sentence could be improved perhaps by stating it as a hypothesis with predictions.

Methods:
General: there is a long list of different methods, some of which are only two sentences long. This makes the presentation of methods a bit difficult to follow. Especially, since some of the methods sections toward the end are not numerically labeled. One possible way to improve this is to clearly state what the purpose of doing each analysis in the methods.
L91: “The study’s methodology …” – verbose. Please revise for conciseness.
L97: List any specific tunings used in the Ultrasonic Cleaning Unit.
L101: “… in order to …” – verbose. Pease revise for conciseness.
L102: “Camel bone samples …” – the use of controls in this sentence needs to be better explained.
L115: “… samples were manually ground …” – please list the conditions of how this took place. Was this done in a sterile environment, what type of mortar/pestle, what temperature, etc.
L135: “… a 100 m diameter …” – is it truly 100 meters?
L165: “In Falcon tubes with 10 mL of the ready SBF, samples were put.” – Please revise the grammar in this sentence.
L216: “… wavelength of 490 nm. was Analyses were performed …” – Please revise this sentence fragment for clarity.

Results:
General: Please ensure that all results sections do not merely repeat parts of the methods.
L243: “After this dentine samples were soaked in a …” - Please revise the grammar in this sentence.
L248: There is no reference to compare the magnitude of change. Remove the adjective “significant”.
L253–255: This section needs to be better explained including the findings.
L270: “… vibrational modes is also found …” – fix tense.
L280: first sentence here is uninformative. Please revise.
L301: section here is uninformative. Please revise.

Discussion:
General: The discussion and conclusion sections generally seem to be missing a bit of direction and a broad perspective with a more thorough review of the literature. One possible way to improve this is to improve the clarity of the aim of the study, then focus the discussion on how this aim was met and borrow perspective from other studies to broaden the context.
L343: “… this study was to study was to develop …” – fix grammatical error.
L346: “… Ratnayake et al. [10].” – fix incorrect reference formatting.
L414: “Limitations of this experiment included …” – Please clarify which experiment this is referring to.
L435: “One of the reasons for this …” – Sentence seems to be missing a citation.
L437–438: Goub et al. and Deligianni et al. – references here seem to be incorrectly formatted.
L444: Gao et al. – reference here seems to be incorrectly formatted.

---

## Round 0.2 · accepted · Accept

Although reviewer indicated that some minor edits are needed, I think that they can be done at the proof stage. Therefore I am accepting your manuscript in its present form.

Reviewer 1 ·

Basic reporting

The article meets the standards and requirements.
All the figures and raw data are as per requirement.
The English language is very professional.

Experimental design

The experimental methods and procedure described are as per the article's requirements.
Clear to understand and all the investigations performed are accurate.

Validity of the findings

The results quoted are valid and accurate.

Additional comments

Below spellings to be corrected.

1. Line 259 – “removed majority “ to be rephrased as “ removed the majority“
2. Line 262 – “indicted “ spell to be corrected “indicated”
3. Line 458 – “ proliferating “ to be rephrased as “ proliferation “